# Influence of Modified Urea Compounds to Improve Nitrogen Use Efficiency under Corn Growth System

Samar Swify [1,2,*], Dovile Avizienyte [1], Romas Mazeika [1] and Zita Braziene [1]

1 Lithuanian Research Centre for Agriculture and Forestry, Instituto Al.1, LT-58344 Akademija, Lithuania
2 Soil and Water Department, Faculty of Agriculture, New Valley University, El-Kharga 72511, Egypt
* Correspondence: samar.swify@lammc.lt

**Abstract:** Adopting new practices is an imperative need to increase the efficiency of nitrogen use (NUE), especially in selecting appropriate N-fertilizer sources and application doses. Regretfully, conventional urea's ability to supply nitrogen to soils is quickly lost as a result of volatilization, leaching, and denitrification. Thus, this study's main aim was to use various modified urea compounds with different doses and investigate their effect on mineral nitrogen release in the soil to improve nitrogen uptake and its use efficiency under the corn growth system. The field trial was conducted in a randomized complete block design (RCBD) by 28 experimental plots. Seven treatments including a control (C), urea (U100 and U200), urea + potassium humate (UPH100 and UPH200), and urea cocrystal (UC100 and UC200) with four replicates were used. The results reported that the treatments significantly ($p < 0.05$) affected grain yields. The urea treatments (U100 = 100 kg N ha$^{-1}$, U200 = 200 kg N ha$^{-1}$) increased the grain yields by 7.16% and 30.53%, respectively, compared to the control (C), while the urea + potassium humate treatments (UPH100 = 100 kg N ha$^{-1}$, UPH200 = 200 kg N ha$^{-1}$) and urea cocrystal treatments (UC100 = 100 kg N ha$^{-1}$, UC200 = 200 kg N ha$^{-1}$) provided a 30.51, 50.47, 39.23, and 56.63% increase in grain yields, respectively, compared to the control. The treatments had significant ($p \leq 0.05$) effects on the fresh leaves and stems yield and the dry matter, fresh cob, and dry cob yields. Moreover, the use of modified urea as urea + potassium humate and urea cocrystal at high rates of 200 kg N ha$^{-1}$ showed highly significant ($p < 001$) effects on the uptake in grain, stems, and total nitrogen uptake by corn compared to the control and urea alone. This study highlighted that modified urea fertilizers such as urea + potassium humate and urea cocrystal were better than conventional urea to improve corn yield productivity and N use efficiency.

**Keywords:** modified urea; nitrogen losses; cocrystal; nitrogen use efficiency; potassium humate

## 1. Introduction

Nitrogen (N) is an essential nutrient of utmost importance in plant nutrition and improving yield. Due to it being more responsible for plant biomass than other mineral nutrients, it is an indispensable component; additionally, it is the driver of plant growth and constitutes 1 to 4% of the dry matter of plants [1,2]. However, nutrition specialists face many problems in adjusting plant requirements and the soil stock of nitrogen. Thus, N fertilizers using have been suggested to enhance crop productivity [3,4]. One of the most significant determinants of crop productivity are nitrogenous fertilizers such as urea (46% N), which is extensively used in the agricultural production sector [5,6]. Forevermore, urea is a common fertilizer that is used due to its relative inexpensiveness, simple handling, and lower production costs compared to other N fertilizers [7]. Urea fertilizers have high solubility when applied to the soil; because of its high solubility, urea-nitrogen may be lost easily from the soil-plant system or become unavailable to plants. Urea loses nitrogen through many processes such as leaching, denitrification, immobilization, and fixation in soil solids as the NH4-N form [8,9]. Nitrogen loss is a vital problem, mainly when used in high-pH soils or low-CEC [10–12]. Nitrogen loss causes urea to be less effective than

other N fertilizers such as ammonium nitrate (AN). A high potential environmental cost is associated with N losses via NH3 volatilization, $NO_3^-$ leaching, and N2O emission to the atmosphere [13,14]. Moreover, plant nitrogen recovery from soluble fertilizers such as urea is reduced to approximately 30–40% [15]. As a result, immediate action is required in order to understand the factors influencing nitrogen availability and to select management practices that minimize nitrogen losses. An adopted suitable strategy will increase the amount of applied N recovered by the crop, improve production efficiency, and reduce the potential environmental impacts of N use [14,16]. A controlled or slow release is one of the most common attempts to reduce nitrogen fertilizer losses by making it in a controlled form, and it is among the most effective N management practices to improve nitrogen use efficiency (NUE) [16,17]. Many studies have been reported that apply to various plant species under different environmental conditions to improve nitrogen use efficiency [10,13,18–29]. The products can be coated, chemically and biochemically modified, or granularly modified [30–35]. Their combination can form stable chemical bonds, reduce the nitrogen release rate, and increase the crop fertilizer use efficiency [36–40]. The application of HA-N significantly raises crop yield, promotes nitrogen absorption and accumulation by crops, and increases nitrogen use efficiency (NUE) [21,36,39,40]. The purpose of this study was to test the hypothesis that modified urea fertilizers containing calcium sulfate and potassium humate could improve mineral nitrogen release and hence its uptake and efficiency as well as maize yield. Therefore, the main objectives of this study were: (i) to investigate the effects of using different modified urea compounds as a new fertilizer on soil mineral nitrogen kinetics, (ii) to enhance the potential to reduce urea nitrogen losses by using different modified urea compounds, (iii) to increase nitrogen uptake and improve nitrogen use efficiency, and (iv) to improve corn grain yield and quality.

## 2. Materials and Methods

### 2.1. Site Description and Experiment Design

This study was carried out in May 2020 in west south Lithuania at Rumokai experimental station of Lithuanian Research Centre for Agriculture and Forestry 54°43′15.7044″ N, 22°58′36.6672″ E to investigate different modified urea products on mineral nitrogen release and its use efficiency under the corn growth system as the test crop. The field soil type was Hapli-Epihypogleyic Luvisol (LVg-p-w-ha) [41] with a moderately silty texture and frequent leaching. The soil chemical properties at 0–20 cm and nitrogen forms concentrations at the soil surface (0–30 cm) and subsurface (0–60 cm) layers are shown in Table 1.

**Table 1.** The chemical characteristics of field soil before using different urea compound fertilizers.

| Soil Properties | Unit | Value | Depth (cm) |
|---|---|---|---|
| $pH_{KCL}$ | - | 6.8 | 0–20 |
| $P_2O_5$ | mg kg$^{-1}$ | 342 | 0–20 |
| $K_2O$ | mg kg$^{-1}$ | 219 | 0–20 |
| SOC | % | 1.48 | 0–20 |
| Nitrogen Mineral | mg kg$^{-1}$ | 27.14 | 0–30 |
| | | 15.68 | 30–60 |
| $NO_3$-N | mg kg$^{-1}$ | 25.64 | 0–30 |
| | | 14.80 | 30–60 |
| $NH_4$-N | mg kg$^{-1}$ | 1.50 | 0–30 |
| | | 0.88 | 30–60 |

SOC = soil organic carbon.

The trial plots were planted with a maize variety, 'Ramirez FAO 160', with a seed ratio of 100,000 pcs. ha$^{-1}$ with 50 cm row spacing and 20 cm seed spacing. The maize was grown without pesticides in a completely randomized block (RCBD) with an area of 4 m$^2$ (2 m × 2 m). The experiment was performed in seven treatments with four replicates and 28 experimental plots. Treatments consisted of control (C) = non-treated,

U100 = 100 kg N ha$^{-1}$ urea, U200 = 200 kg N ha$^{-1}$ urea, UPH100 = 100 kg N ha$^{-1}$ urea + potassium humate (UPH), UPH200 = 200 kg N ha$^{-1}$ urea + potassium humate (UPH), UC100 = 100 kg N ha$^{-1}$ urea cocrystal (UC) and UC200 = 200 kg N ha$^{-1}$ urea cocrystal (UC). The fertilizers were manually applied in accordance with the experimental scheme. The corn productivity and biometrics were measured manually during the corn physiological maturity phase in September 2020 (BBCH 88-89). After harvesting, the remainder of the plant remains were plowed at a depth of 22 cm. Next, 28 plants were randomly selected for grain yield and grain quality determination. For dry biomass and yield weight determinations, samples were taken from all the replicates and oven-dried at 65 ± 5 °C until constant weight.

### 2.2. Soil Sampling and Analytical Procedures

Three replicates of soil samples were collected at 0–30 and 30–60 cm deep in untreated and treated plots. The samples were taken using a stainless-steel push tube with three sub-samples per composite plot to form a sample. All soil samples were air-dried and crushed into a sieve with 2 mm openings. The Agrochemical Research Laboratory analyzed the soil and mineral nitrogen properties at the Lithuanian Agricultural and Forest Research Centre [42]. Soil mobile potassium as $K_2O$ and phosphorus as $P_2O_5$ were extracted using 1:20 (wt./vol) soil suspension of ammonium lactate–acetic acid extractant (pH 3.7). The suspension was shaken for 4 h. Mobile $P_2O_5$ was extracted using ammonium molybdate via the spectrometric method with Shimadzu UV 1800 spectrophotometer, while mobile $K_2O$ was determined using flame emission spectroscopy with flame emission spectroscopy JENWAY PFP7 flame photometer. The soil pH was determined with a 1:5 (vol/vol) soil suspension in 1M KCl. The mix was shaken for 60 min and allowed to sit for 1 h. The pH of the suspension was measured at 20 ± 2 °C while stirring with a pH meter. Mineral nitrogen was extracted in 1:5 (wt./vol) soil suspension of 1 M KCl solution. The suspension was shaken for 60 min at 20 ± 2 °C. After shaking, the suspension was filtrated and analyzed using a flow injection analysis (FIA) system by FIASTAR 5000 analyzer. After the addition of an acidic sulfanilamide solution, the nitrates in the soil extract were converted to nitrites in the cadmium column. They then reacted with N-(1-naphthyl) ethylenediamine dihydrochloride to form a purple azo dye whose absorbance can be measured at 540 nm and 720 nm. The sum of nitrate and nitrite nitrogen was determined in the soil extract. The soil extract was injected into a flowing carrier solution, where ammonium was mixed with sodium hydroxide to form gaseous ammonia, which passed through a gas-permeable membrane into the indicator stream. Acidic indicators change color in this stream when they react with ammonium gas. Photometric measurements were then performed at 540 nm and 720 nm. Calculating the mineral nitrogen involves adding the combined amounts of nitrate and nitrite nitrogen to the ammonia nitrogen. The organic carbon content was determined using dry combustion, where the sample was heated to 900 °C in a stream of air, and the carbon dioxide formed was measured using infrared spectroscopy.

### 2.3. Fertilizer Materials

The study was conducted by comparing urea as a mineral nitrogen fertilizer with other modified urea compound fertilizers such as urea + potassium humate (UPH) and urea cocrystal (UC):

1. Urea (Total N 46.2%).
2. Urea + potassium humate (UPH) contains 40% N and 1% potassium humate; produced by specialists of AB Achema scientific experimental laboratory by coating urea granules with potassium humate.
3. Urea cocrystal ($4urea.CaSO_4$) contains 29.8% N, 8.5% S, and 10.6% Ca. The fertilizer was mechanochemically produced by chemists from Kaunas Technology University (KTU), Faculty of Chemical Technology, Lithuania.

### 2.4. Metrological Conditions

The annual average precipitation during the period of 1982–2010 ranged from 32 to 81 mm, with a high average in August, and an annual average temperature of 15.38 °C. The area has a medium temperate climate with an average temperature of 16.23 °C during the maize-growing season in 2020, with the high-temperature values recorded in June and August being an average of 18.63 and 18.87 °C, respectively. The annual precipitation was 23.62 mm in 2020, which is less than the long-term average (Figure 1).

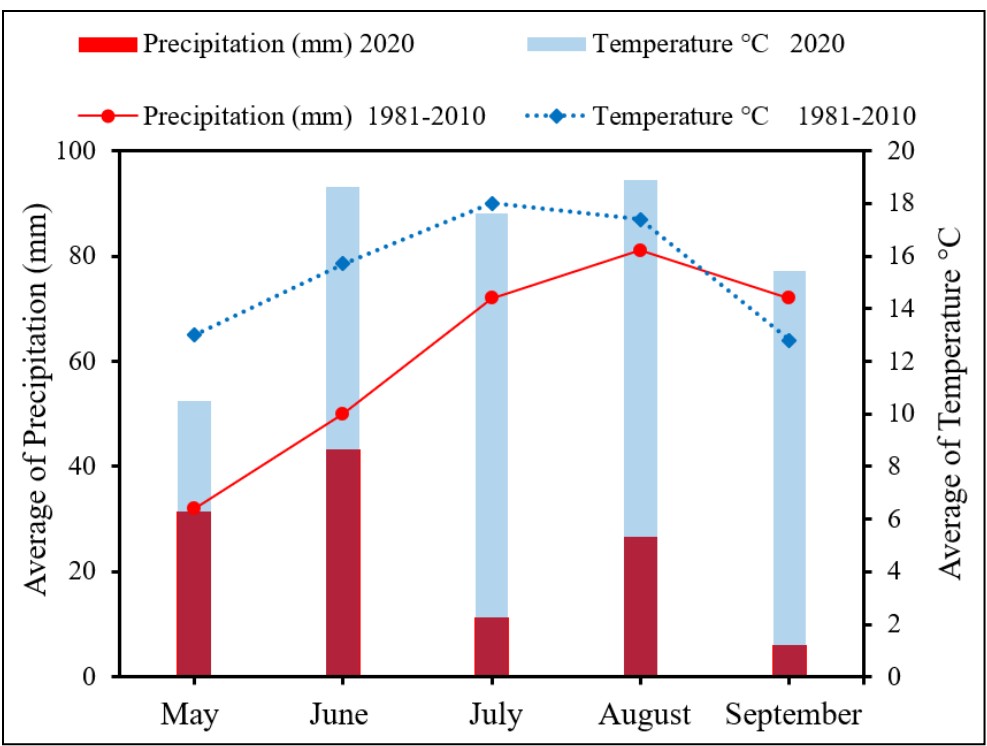

**Figure 1.** The distribution of monthly rainfall (mm) and temperature at the experimental site during 2020 and average rainfall and temperature for the past 30 years at the site.

### 2.5. Nitrogen Uptake and Use Efficiency Estimation

The term refers to nitrogen uptake and agronomic nitrogen use efficiency (ANUE) expressions as follows [28]:

$$N\ uptake = N\%\ in\ grain\ or\ stems \times dry\ matter\ of\ grain\ or\ stems\ in\ \left(kg\ ha^{-1}\right) \tag{1}$$

$$ANUE\left(kg\ grain\ increased\ kg\ N^{-1}\ applied\right) = \frac{YN - Y0}{NR} \tag{2}$$

In the above expressions, *YN* and *Y0* are the yields (kg ha$^{-1}$) in the fertilized and control (no fertilizer) plots, respectively, and *NR* refers to the rate of fertilizer applied (kg ha$^{-1}$).

### 2.6. Statistical Analysis

Analysis of variance (ANOVA) was performed using a general linear model on plant density, fresh leaves and stems yield, as well as dry matter, fresh and dry ear yields and grain yields, grain characteristics, N uptake, and ANUE. Pearson's correlation shows the relationship between time and mineral N with respect to its forms (nitrate and ammonium). The statistical analysis software was IBM SPSS 25.0, and Duncan's test at the 5% level was performed to separate means according to the ANOVA results.

## 3. Results

### 3.1. Plants' Density, Fresh Leaves and Stems, and Dry Matter Yields

There was a significant treatment effect on the observed corn plants' density. The plants' density ranged from 89,450 plants ha$^{-1}$ in the control to 97,800 plants ha$^{-1}$ in UC200. The treatments showed highly significant effects on the fresh leaves and stems yield and dry matter, as shown in Figure 2. The treatments of UPH200 > UC200 recorded the highest fresh leaves and stems yield with means of 48.05 and 50.27 t/ha$^{-1}$, respectively, followed by the treatment of UC200 > UPH100 with means of 43.61 and 44.72 t/ha$^{-1}$, respectively.

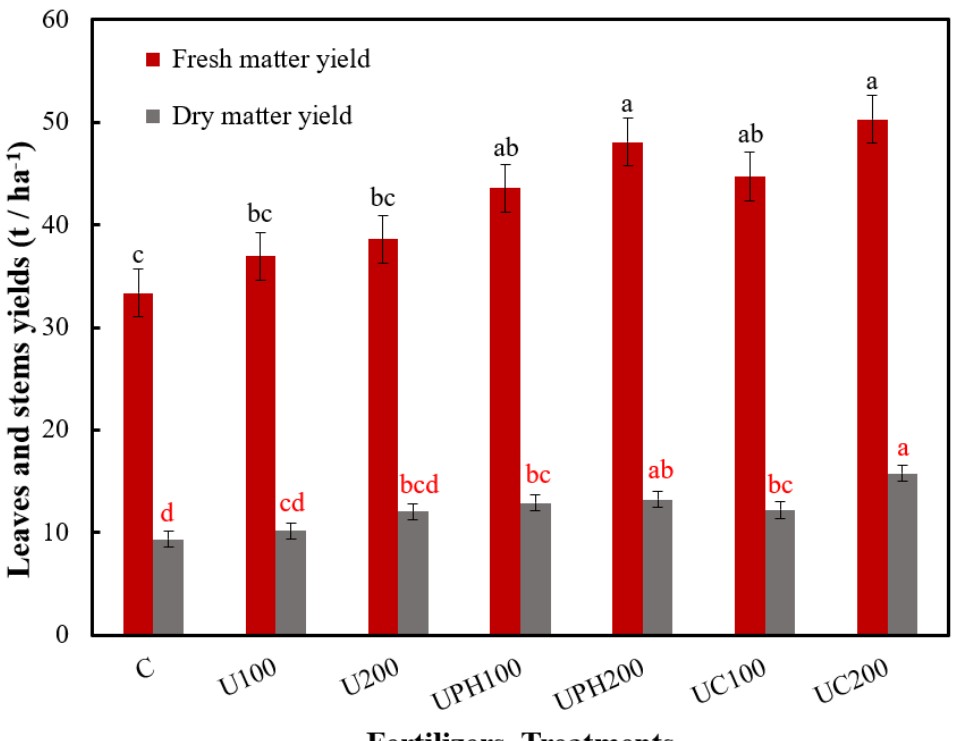

**Figure 2.** Fresh leaves and stems and dry matter yields followed by Duncan's multiple range test letters at 5% level. The letters with the same color have the same significance level test. Note. Control (C) = non-treated, U100 = 100 kg N ha$^{-1}$ urea, U200 = 200 kg N ha$^{-1}$ urea, UPH100 = 100 kg N ha$^{-1}$ urea + potassium humate (UPH), UPH200 = 200 kg N ha$^{-1}$ urea + potassium humate (UPH), UC100 = 100 kg N ha$^{-1}$ urea cocrystal (4urea.CaSO$_4$), and UC200 = 200 kg N ha$^{-1}$, urea cocrystal (4urea.CaSO$_4$).

There were no significant differences recorded for the urea treatments U100 and U200 compared to the control, which recorded the lowest fresh leaves and stems weight among the fertilizer treatments with means of 36.94 and 38.61 t/ha$^{-1}$, respectively. The dry matter yields ranged from 9.34 t/ha$^{-1}$ in the control to 15.75 t/ha$^{-1}$ in UC200 (Table 2). The treatment had significant effects on fresh and dry matter yields compared to the control except for the urea treatments (U100 and U200), which exhibited nonsignificant effects on the control. The urea cocrystal treatments UC100 and UC200 recorded the highest fresh silage yields and dry matter with means of 44.72, 50.27 t/ha$^{-1}$, 12.19, and 15.75 t/ha$^{-1}$, respectively.

**Table 2.** Means (±standard deviation) of plants' density, silage dry matter, and fresh silage yields after using different urea compound fertilizers followed by Duncan's multiple range test letters at the 5% level.

| Treatments | Plant Density | Fresh Leaves and Stems Yield | Dry Matter | % Moisture |
|---|---|---|---|---|
| | 1000 plants ha$^{-1}$ | t/ha$^{-1}$ | | |
| C | 89.450 ± 2.6 c | 33.33 ± 6.3 c | 9.34 ± 1.3 d | 71.97 |
| U100 | 90.825 ± 4.7 c | 36.94 ± 6.7 bc | 10.15 ± 1.6 cd | 72.52 |
| U200 | 92.325 ± 2.3 bc | 38.61 ± 7.4 bc | 12.03 ± 2.3 bcd | 68.84 |
| UPH100 | 93.350 ± 5.1 abc | 43.61 ± 4.7 ab | 12.88 ± 1.6 bc | 70.46 |
| UPH200 | 96.675 ± 2.3 ab | 48.05 ± 6.4 a | 13.23 ± 2.0 ab | 80.88 |
| UC100 | 96.675 ± 2.3 ab | 44.72 ± 3.8 ab | 12.19 ± 1.2 bc | 72.74 |
| UC200 | 97.800 ± 0.0 a | 50.27 ± 5.1 a | 15.75 ± 1.7 a | 68.66 |
| *p*-value | *p* < 0.011 | *p* < 0.006 | *p* < 0.002 | - |

Note 1. Control (C) = non-treated, U100 = 100 kg N ha$^{-1}$ urea, U200 = 200 kg N ha$^{-1}$ urea, UPH100 = 100 kg N ha$^{-1}$ urea + potassium humate (UPH), UPH200 = 200 kg N ha$^{-1}$ urea + potassium humate (UPH), UC100 = 100 kg N ha$^{-1}$ urea cocrystal (4urea.CaSO$_4$), and UC200 = 200 kg N ha$^{-1}$, urea cocrystal (4urea.CaSO$_4$). Note 2. Values in the same column followed by the same letter are not different (*p* < 0.05) according to Duncan's multiple range test at the 5% level.

### 3.2. Grain Yields and Grain Quality

The mean grain yields followed by Duncan's test letters are shown in Figure 3. There were highly significant differences between the effects of the treatments on grain yields. The mean grain yields ranged from 12.84 t/ha$^{-1}$ in the control to 20.11 t/ha$^{-1}$ in UC200, as shown in Table 3. The urea treatments (U100, U200) increased grain yields by 7.16% and 30.53%, respectively, compared to the control (C), while the urea + potassium humate (UPH100, UPH200) and urea cocrystal treatments (UC100, UC200) provided a 30.51, 50.47, 39.23, and 56.63% increase in grain yields, respectively, compared to the control. Additionally, the treatments had significant effects on both the fresh and dry cob yields. The mean fresh cob yields ranged from 26.11 t/ha$^{-1}$ in the control to 36.95 t/ha$^{-1}$ in UC200, and the mean dry cob yields ranged from 18.69 t/ha$^{-1}$ in the control to 25.88 t/ha$^{-1}$ in UC200 (Table 3). Compared to the control, the modified urea treatments recorded the highest grain and cob yields, as well as fresh and dry matter yields (UC200 > UPH200 > UC100).

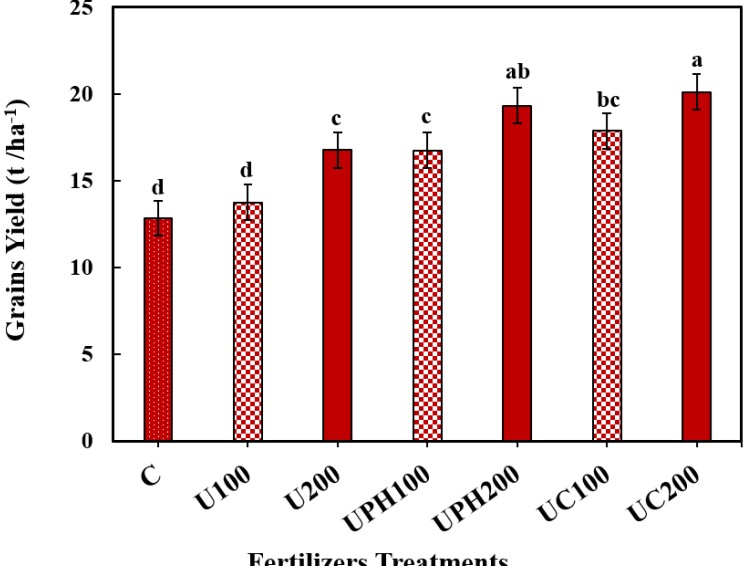

**Figure 3.** The corn grain yield means (t/ha$^{-1}$) of the field experiment after using different urea compounds fertilizers followed by Duncan's multiple range test letters at 5% level. Note. Control (C)

= non-treated, U100 = 100 kg N ha$^{-1}$ urea, U200 = 200 kg N ha$^{-1}$ urea, UPH100 = 100 kg N ha$^{-1}$ urea + potassium humate (UPH), UPH200 = 200 kg N ha$^{-1}$ urea + potassium humate (UPH), UC100 = 100 kg N ha$^{-1}$ urea cocrystal (4urea.CaSO$_4$), and UC200 = 200 kg N ha$^{-1}$, urea cocrystal (4urea.CaSO$_4$).

**Table 3.** Means (±standard deviation) of grain yields and fresh and dry cob yields after using different urea compound fertilizers followed by Duncan's test letters.

| Treatments | Grain Yields | Fresh Ears Yield | Dry Ears Yield | % Moisture of Ears Yields |
|---|---|---|---|---|
| | | t/ha$^{-1}$ | | |
| C | 12.84 ± 1.3 d | 26.11 ± 4.7 c | 18.69 ± 3.1 c | 28.42 |
| U100 | 13.75 ± 1.7 d | 28.89 ± 6.3 bc | 20.21 ± 4.5 bc | 30.04 |
| U200 | 16.76 ± 1.0 c | 29.45 ± 3.2 bc | 21.16 ± 1.4 bc | 28.14 |
| UPH100 | 16.76 ± 0.7 c | 32.22 ± 2.2 abc | 22.31 ± 1.4 bc | 30.75 |
| UPH200 | 19.32 ± 0.5 ab | 33.62 ± 4.7 ab | 23.55 ± 2.5 ab | 29.95 |
| UC100 | 17.88 ± 2.5 bc | 33.33 ± 4.8 ab | 22.18 ± 4.6 abc | 33.45 |
| UC200 | 20.11 ± 1.1 a | 36.95 ± 2.5 a | 25.88 ± 1.7 a | 29.95 |
| *p*-value | *p* < 0.001 | *p* < 0.025 | *p* < 0.043 | - |

Note 1. Control (C) = non-treated, U100 = 100 kg N ha$^{-1}$ urea, U200 = 200 kg N ha$^{-1}$ urea, UPH100 = 100 kg N ha$^{-1}$ urea + potassium humate (UPH), UPH200 = 200 kg N ha$^{-1}$ urea + potassium humate (UPH), UC100 = 100 kg N ha$^{-1}$ urea cocrystal (4urea.CaSO$_4$), and UC200 = 200 kg N ha$^{-1}$, urea cocrystal (4urea.CaSO$_4$). Note 2. Values in the same column followed by the same letter are not different (*p* < 0.05) according to Duncan's multiple range test at the 5% level.

Table 4 shows that the treatments had significant effects on the characteristics of the grains, including grain protein, total N, and grain carbohydrate content, but did not affect the grain dry matter. The grain protein means ranged from 6.21% in the control to 7.67% in UC200. The treatments of U100 and UPH100 recorded the lowest total N in the grains by 1.14% after the control. In contrast, UC200 had the highest total N concentration at 1.23%. The grain dry matter ranged from 93.68% to 95.13% in UPH200 and UPH100, respectively. The treatments had highly significant effects on the grain carbohydrate content. It ranged from 70.51% in the control to 76.69% in UC200.

**Table 4.** Means (±standard deviation) of grain characteristics (protein, total N, dry matter, and carbohydrates) after using different Urea compounds fertilizers.

| Treatments | % Protein | % Total N | % Dry Matter | % Carbohydrates |
|---|---|---|---|---|
| C | 6.21 ± 0.2 b | 0.99 ± 0.0 b | 94.02 ± 1.2 | 70.51 ± 0.8 d |
| U100 | 7.16 ± 0.4 a | 1.14 ± 0.1 a | 94.35 ± 1.4 | 70.77 ±0.2 d |
| U200 | 7.19 ± 0.7 a | 1.18 ± 0.0 a | 94.56 ± 0.9 | 72.39 ±0.2 c |
| UPH100 | 7.11 ± 0.4 a | 1.14 ± 0.1 a | 95.13 ± 1.0 | 72.85± 0.1 bc |
| UPH200 | 7.39 ± 0.7 a | 1.18 ± 0.1 a | 93.68 ± 0.8 | 72.22± 0.8 c |
| UC100 | 7.29 ± 0.5 a | 1.17 ± 0.1 a | 93.82 ± 1.3 | 73.43 ± 0.8 b |
| UC200 | 7.67 ± 0.2 a | 1.23 ± 0.0 a | 94.51 ± 1.3 | 76.69 ±0.8 a |
| *p*-value | *p* < 0.007 | *p* < 0.001 | ns | *p* < 0.001 |

Note 1. Control (C) = non-treated, U100 = 100 kg N h$^{-1}$ urea, U200 = 200 kg N h$^{-1}$ urea, UPH100 = 100 kg N h$^{-1}$ urea + potassium humate (UPH), UPH200 = 200 kg Nh$^{-1}$ urea + potassium humate (UPH), UC100 = 100 kg N h$^{-1}$ urea cocrystal (4urea.CaSO$_4$), and UC200 = 200 kg N h$^{-1}$, urea cocrystal (4urea.CaSO$_4$). Note 2. Values in the same column followed by the same letter are not different (*p* < 0.05) according to Duncan's multiple range test at the 5% level.

### 3.3. Mineral Nitrogen Release

Some pre-studies examined the release process of mineral nitrogen and its forms ($NO_3^-$ and $NH_4^+$) from urea compound granules in the soil without plants (Figure 4). The process was divided into three stages including (1) low concentration, (2) increased concentration and (3) stability of concentration. Additionally, the results showed that the concentration of ammonium, nitrate, and mineral nitrogen increased with time due to the decomposition of urea compound granules. The relation between the transformation of

soil mineral nitrogen forms depended on soil water content and urea compound type. The mineral nitrogen concentration started with a low concentration in the beginning until three days, then increased to a maximum over two weeks. After that, the nitrogen concentration remained somewhat stable or decreased depending on the microbial activity or ammonia volatilization.

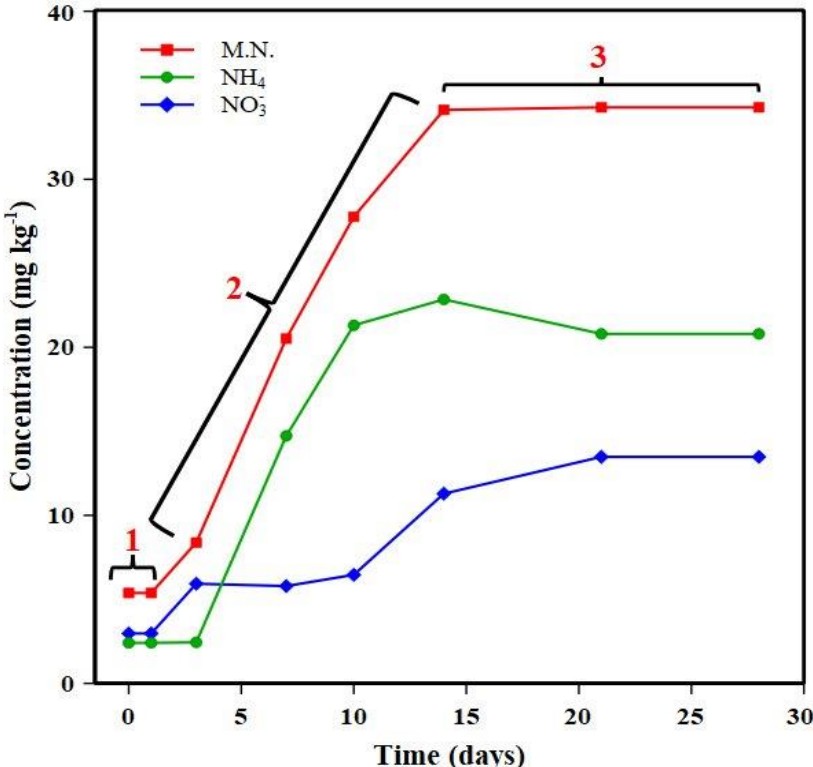

**Figure 4.** The release process of mineral nitrogen from urea compound granules as a relation between the concentration of mineral nitrogen forms ($NO_3^-$ and $NH_4^+$) and time.

As shown in Table 5, the mineral N in the soil surface layer (0–30 cm) during the corn's growth was significantly correlated with time for all the treatments except U200 and UPH100. Additionally, the soil surface layer (0–30 cm) had a higher mineral nitrogen concentration than the soil subsurface layer (30–60 cm) as shown in Table 5.

However, the concentration correlated highly significantly with all the treatments in the soil subsurface layer (30–60 cm). Approximately 30% of the mineral nitrogen was found in the subsurface layer (30–60 cm) (Figure 5). The control recorded the lowest mineral nitrogen concentration values in the soil surface (0–30 cm) and subsurface (30–60 cm) layers with a mean of 16.05 and 10.72 mg kg$^{-1}$, respectively. The treatment of UC200 recorded the highest means of 29.34 and 13.74 mg kg$^{-1}$. in soil surface (0–30 cm) and subsurface (30–60 cm) layers, respectively. The treatments of U200 and UPH200 showed high means of mineral nitrogen after UC200 in the soil surface (0–30 cm) with values of 27.59 and 26.13 mg kg$^{-1}$, respectively, but in the soil subsurface (30–60 cm)layer, U100 and U200 recorded high concentrations with means of 13.59 and 13.50 mg kg$^{-1}$, respectively, after UC200.

**Table 5.** The relation between the time and mineral nitrogen concentration (mg kg$^{-1}$) in soil surface (0–30 cm) and subsurface (30–60 cm) layers during the corn's growth.

| Treatments | Mean | Pearson Correlation | Sig (2-Tailed) |
| --- | --- | --- | --- |
| depth (0–30 cm) | | | |
| C | 16.05 | −0.820 ** | 0.007 |
| U100 | 23.80 | −0.733 * | 0.025 |
| U200 | 27.59 | −0.666 | 0.050 |
| UPH100 | 25.25 | −0.587 | 0.097 |
| UPH200 | 26.13 | −0.727 * | 0.026 |
| UC100 | 21.72 | −0.711 * | 0.032 |
| UC200 | 29.34 | −0.703 * | 0.035 |
| depth (30–60 cm) | | | |
| C | 10.72 | −0.808 ** | 0.008 |
| U100 | 13.59 | −0.834 ** | 0.005 |
| U200 | 13.50 | −0.814 ** | 0.008 |
| UPH100 | 12.11 | −0.868 ** | 0.002 |
| UPH200 | 12.52 | −0.887 ** | 0.001 |
| UC100 | 11.81 | −0.867 ** | 0.002 |
| UC200 | 13.74 | −0.721 * | 0.028 |

** Correlation is significant at the 0.01 level (two-tailed). * Correlation is significant at the 0.05 level (two-tailed).

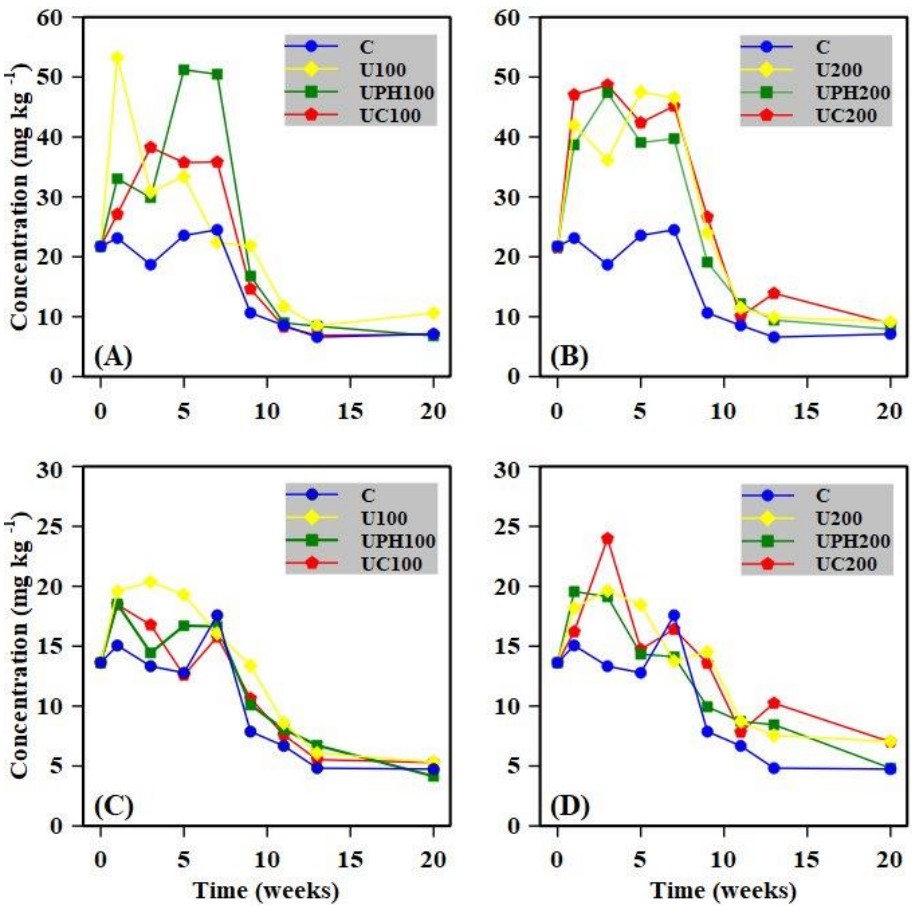

**Figure 5.** The relation between the mineral nitrogen concentration and time during the maize growth period in soil surface 0–30 cm (**A**,**B**) and subsurface 30–60 cm (**C**,**D**). Note. Control (C) = non-treated, U100 = 100 kg N ha$^{-1}$ urea, U200 = 200 kg N ha$^{-1}$ urea, UPH100 = 100 kg N ha$^{-1}$ urea + potassium humate (UPH), UPH200 = 200 kg N ha$^{-1}$ urea + potassium humate (UPH), UC100 = 100 kg N ha$^{-1}$ urea cocrystal (4urea.CaSO$_4$), and UC200 = 200 kg N ha$^{-1}$, urea cocrystal (4urea.CaSO$_4$).

The soil mineral nitrogen formed into ammonium ($N-NH_4^+$) and nitrate ($N-NO_3^-$) in all the different urea compound granules during the release process. The ammonium ($N-NH_4^+$) concentration was lower than the nitrate concentration for all the treatments. The soil surface (0–30 cm) layer recorded a higher ammonium concentration than the soil subsurface (30–60 cm) layer. Before fertilization and corn growth, the ammonium concentration means were 1.54 and 1.17 mg kg$^{-1}$ in the soil surface (0–30 cm) and subsurface (30–60 cm) layers, respectively. The ammonium concentration fluctuated slightly and decreased during the corn growth until harvest (Figure 6). Therefore, the concentration did not correlate with time during the corn growth, as shown in Table 6.

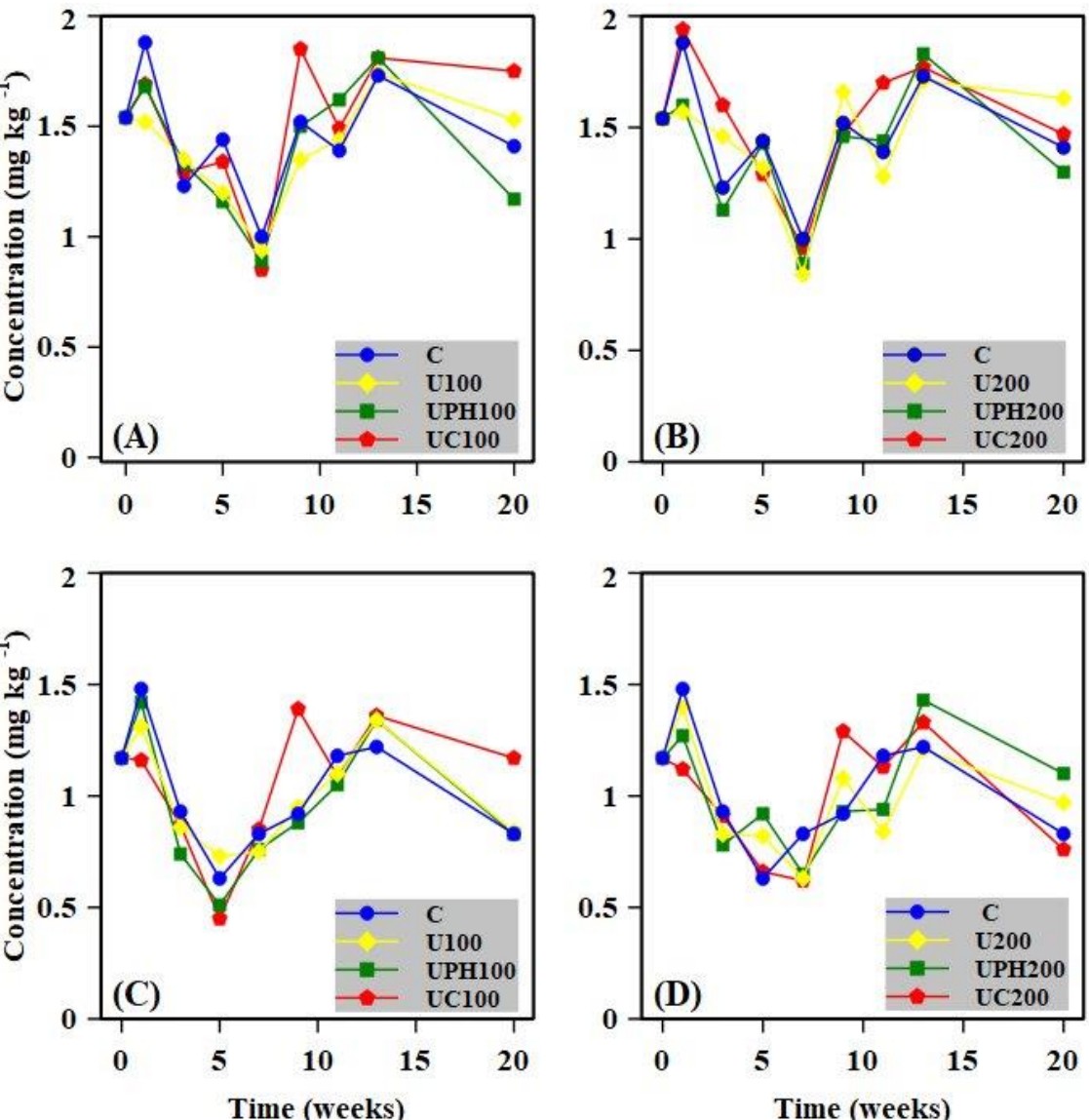

**Figure 6.** The relation between the ammonium concentration and time during the maize growth period in soil surface 0–30 cm (**A**,**B**) and subsurface 30–60 cm (**C**,**D**). Note. Control (C) = non-treated, U100 = 100 kg N ha$^{-1}$ urea, U200 = 200 kg N ha$^{-1}$ urea, UPH100 = 100 kg N ha$^{-1}$ urea + potassium humate (UPH), UPH200 = 200 kg N ha$^{-1}$ urea + potassium humate (UPH), UC100 = 100 kg N ha$^{-1}$ urea cocrystal (4urea.CaSO$_4$), and UC200 = 200 kg N ha$^{-1}$, urea cocrystal (4urea.CaSO$_4$).

**Table 6.** The relation between time and ammonium concentration (mg kg$^{-1}$) in soil surface (0–30 cm) and subsurface (30–60 cm) layers during the corn's growth.

| Treatments | Means | Pearson Correlation | Sig (2-Tailed) |
|---|---|---|---|
| | | depth (0–30 cm) | |
| C | 1.46 | −0.100 | 0.799 |
| U100 | 1.40 | 0.228 | 0.555 |
| U200 | 1.45 | 0.184 | 0.636 |
| UPH100 | 1.41 | −0.094 | 0.810 |
| UPH200 | 1.40 | 0.028 | 0.943 |
| UC100 | 1.51 | 0.335 | 0.379 |
| UC200 | 1.53 | −0.075 | 0.848 |
| | | depth (30–60 cm) | |
| C | 1.02 | −0.264 | 0.493 |
| U100 | 1.01 | −0.145 | 0.710 |
| U200 | 0.99 | −0.173 | 0.656 |
| UPH100 | 0.97 | −0.111 | 0.777 |
| UPH200 | 1.02 | 0.128 | 0.742 |
| UC100 | 1.06 | 0.310 | 0.417 |
| UC200 | 1.00 | −0.066 | 0.867 |

The treatments had no effect on the ammonium concentration in the soil surface layer (0–30 cm), but in the soil subsurface layer (30–60 cm), there were significant differences ($p < 0.003$) between the treatments after one, three, five, and nine weeks, and after the harvest as well. After the harvest, the ammonium recorded a slight decrease with a mean of 1.46 mg kg$^{-1}$ in the soil surface layer (0–30 cm), and the soil subsurface (30–60 cm) layer recorded 0.93 mg kg$^{-1}$ as shown in Figure 6.

The nitrate (N-NO$_3{}^{-}$) had the same trend as the mineral nitrogen, starting with a low concentration before the fertilization, and during corn growth recorded a mean of 20.19 and 12.45 mg kg$^{-1}$ in the soil surface (0–30 cm) and subsurface (30–60 cm) layers, respectively (Figure 7). In the soil surface 0–30 cm layer, the nitrate concentration correlated significantly with time for all the treatments except U200 and UPH100, as shown in Table 7.

**Table 7.** The relation between time and nitrate concentration (mg kg$^{-1}$) in soil surface (0–30 cm) and subsurface (30–60 cm) layers during the corn's growth.

| Treatments | Means | Pearson Correlation | Sig (2-Tailed) |
|---|---|---|---|
| | | depth (0–30 cm) | |
| C | 14.59 | −0.811 ** | 0.008 |
| U100 | 22.39 | −0.734 * | 0.024 |
| U200 | 26.14 | −0.662 | 0.052 |
| UPH100 | 23.84 | −0.579 | 0.102 |
| UPH200 | 24.73 | −0.721 * | 0.028 |
| UC100 | 20.21 | −0.707 * | 0.033 |
| UC200 | 28.36 | −0.694 * | 0.038 |
| | | depth (30–60 cm) | |
| C | 9.70 | −0.791 * | 0.011 |
| U100 | 12.58 | −0.819 ** | 0.007 |
| U200 | 12.50 | −0.803 ** | 0.009 |
| UPH100 | 11.14 | −0.855 ** | 0.003 |
| UPH200 | 11.50 | −0.882 ** | 0.002 |
| UC100 | 10.75 | −0.864 ** | 0.003 |
| UC200 | 12.74 | −0.709 * | 0.033 |

** Correlation is significant at the 0.01 level (two-tailed). * Correlation is significant at the 0.05 level (two-tailed).

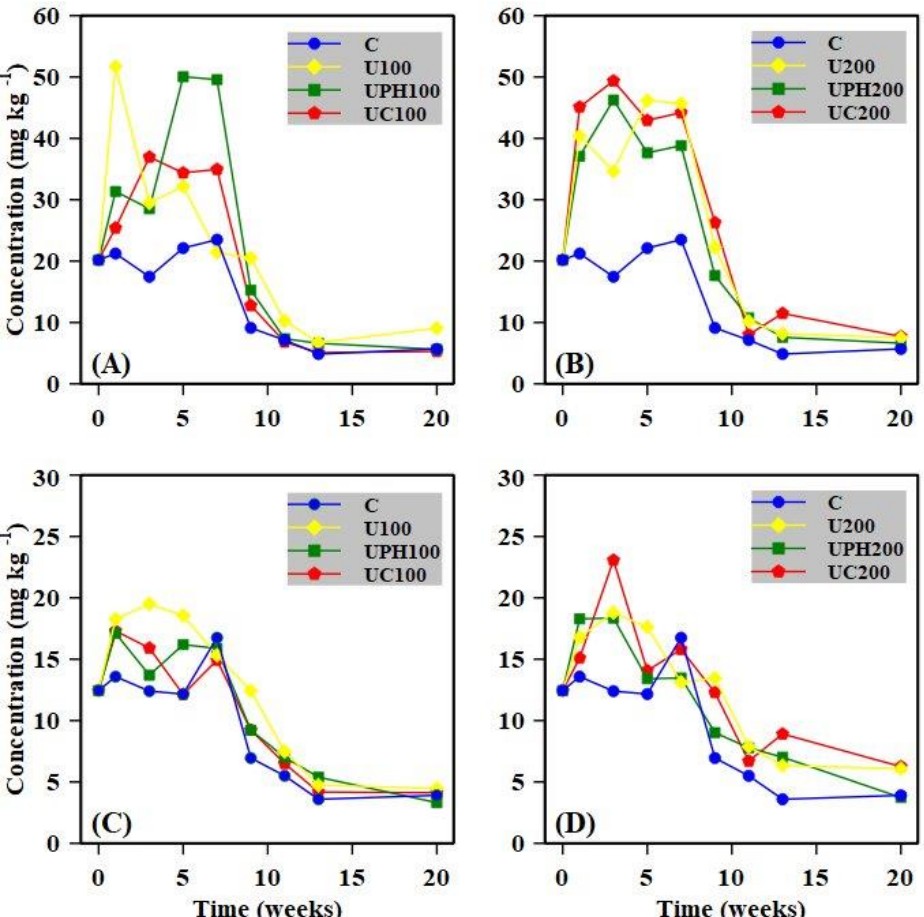

**Figure 7.** The relation between the nitrate concentration during the maize growth period in soil surface 0–30 cm (**A,B**) and subsurface 30–60 cm (**C,D**). Note. Control (C) = non-treated, U100 = 100 kg N ha$^{-1}$ urea, U200 = 200 kg N ha$^{-1}$ urea, UPH100 = 100 kg N ha$^{-1}$ urea + potassium humate (UPH), UPH200 = 200 kg N ha$^{-1}$ urea + potassium humate (UPH), UC100 = 100 kg N ha$^{-1}$ urea cocrystal (4urea.CaSO$_4$), and UC200 = 200 kg N ha$^{-1}$, urea cocrystal (4urea.CaSO$_4$).

The highest concentration was recorded after 5 weeks with a mean of 37.93 mg kg$^{-1}$. The treatments had significant effects on the soil nitrate concentration during all the corn growth periods and after the harvest ($p < 0.007$), except after eleven weeks it was not significant. The treatments of UC200 and U200 recorded the highest means of 28.36 and 26.14 mg kg$^{-1}$, respectively. The soil subsurface (30–60 cm) layer had approximately 25% nitrate concentration during corn growth. Thus, the nitrate concentration was significantly correlated with time for all the treatments in the soil subsurface layer (30–60 cm) (Table 7). The treatments significantly affected the nitrate concentration until the fifth week and after thirteen weeks with $p < 0.004$. The concentration after seven weeks started to decrease until the harvest for all treatments in the soil surface 0–30 cm and soil subsurface layers (30–60 cm). After the harvest, the concentration was 6.78 and 4.55 mg kg$^{-1}$ in the soil surface (0–30 cm) and soil subsurface (30–60 cm) layers, respectively (Figure 7).

*3.4. Nitrogen Uptake and Use Efficiency*

The statistics results in Table 8 showed that the use of modified urea as urea + potassium humate and urea cocrystal with high rates of 200 kg N ha$^{-1}$ had highly significant effects on the N uptake in grain and stems and the total nitrogen uptake by the corn crop compared to the control and urea alone (Figure 8). In addition, the treatments of UPH200 and UC200 provided an increase of 79.15 and 93.677%, respectively, in grain N uptake compared to the control, and it recorded the highest N uptake in the stems with means of

133.58 and 168.49 kg ha$^{-1}$, respectively. The treatments significantly affected the agronomic nitrogen use efficiency (ANUE). Urea + potassium humate and urea cocrystal improved the nitrogen use efficiency, recording means of 50.40, 39.20, 36.38, and 32.41 kg kg$^{-1}$ at UC100 > UPH100 > UC200 > UHP200, while urea alone recorded the lowest means of 9.20 and 19.61 kg kg$^{-1}$ at U200 > U100 for agronomic nitrogen use efficiency (ANUE) as shown in Figure 8.

**Table 8.** Effect of modified urea fertilizers on grain and stem N uptake, and agronomic N use efficiency (ANUE) in the corn crops.

| Treatments | N Uptake in Grains | N Uptake in Stems | N Uptake [Grain + Stems] | ANUE |
|---|---|---|---|---|
| | \multicolumn{3}{}{kg ha$^{-1}$} | | kg kg$^{-1}$ |
| C | 127.62 e | 73.79 d | 201.41 e | - |
| U100 | 156.87 d | 104.58 c | 261.45 d | 9.20 d |
| U200 | 199.16 bc | 120.27 bc | 319.43 bc | 19.61 cd |
| UPH100 | 190.90 c | 101.74 c | 292.63 cd | 39.20 ab |
| UPH200 | 228.64 ab | 133.58 b | 362.23 b | 32.41 bc |
| UC100 | 209.09 bc | 96.35 cd | 305.44 dc | 50.40 a |
| UC200 | 247.17 a | 168.49 a | 415.66 a | 36.38 ab |
| SE± | 8.09 | 6.09 | 13.42 | 3.53 |
| *p*-value | *p* < 0.001 | *p* < 0.001 | *p* < 0.001 | *p* < 0.001 |

Note 1. Control (C) = non-treated, U100 = 100 kg N h$^{-1}$ urea, U200 = 200 kg N h$^{-1}$ urea, UPH100 = 100 kg N h$^{-1}$ urea + potassium humate (UPH), UPH200 = 200 kg N h$^{-1}$ urea + potassium humate (UPH), UC100 = 100 kg N h$^{-1}$ urea cocrystal (4urea.CaSO$_4$), and UC200 = 200 kg N h$^{-1}$, urea cocrystal (4urea.CaSO$_4$). Note 2. Values in the same column followed by the same letter are not different (*p* < 0.05) according to Duncan's multiple range test at the 5% level.

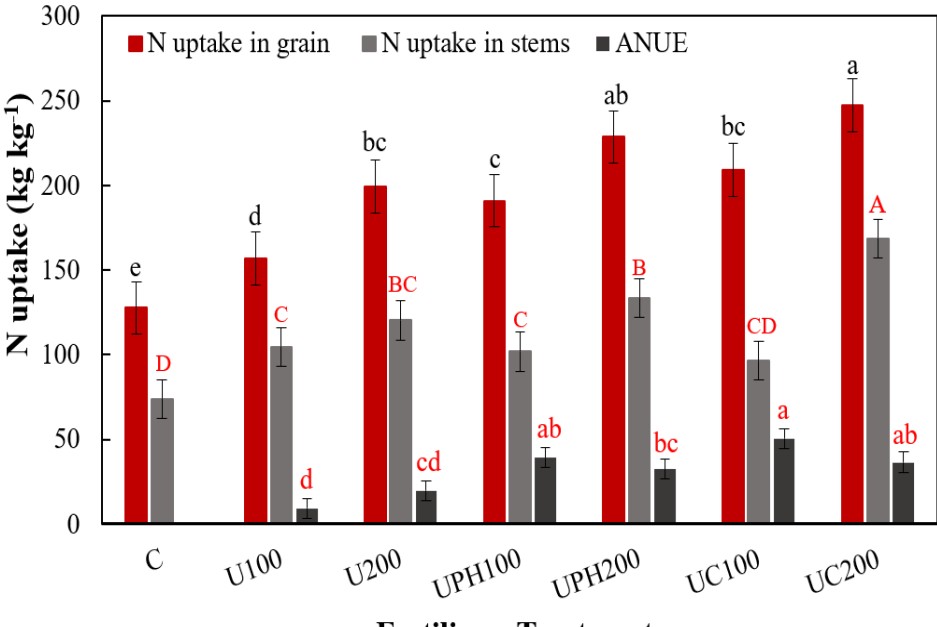

**Figure 8.** The effect of urea compound treatments on N uptake (kg kg$^{-1}$) in maize grains and stems, and ANUE (kg kg$^{-1}$) followed by Duncan's multiple range test letters at 5% level. The letters with the same color have the same significance level test. Note. Control (C) = non-treated, U100 = 100 kg N ha$^{-1}$ urea, U200 = 200 kg N ha$^{-1}$ urea, UPH100 = 100 kg N ha$^{-1}$ urea + potassium humate (UPH), UPH200 = 200 kg N ha$^{-1}$ urea + potassium humate (UPH), UC100 = 100 kg N ha$^{-1}$ urea cocrystal (4urea.CaSO$_4$), and UC200 = 200 kg N ha$^{-1}$, urea cocrystal (4urea.CaSO$_4$).

## 4. Discussion

### 4.1. Effect of Treatments on Fresh and Dry Silage Yields, Grain Yields, and Grain Characteristics

In the present study, the application of modified urea compounds provided a consistent improvement in grain yields and grain quality compared with conventional urea treatments. Moreover, the results were mostly supported by many previous studies that reported that the use of urea alone could increase nitrogen losses by more than 15% under the current climate change [16,17,43]. Furthermore, the N recovery by crops from soluble N fertilizers such as urea is often as low as 30–40%, with a potentially high environmental cost associated with N losses via $NH_3$ volatilization [32,44]. Therefore, it was necessary to use some compounds that help reduce urea N losses and improve the efficiency of using urea as an essential fertilizer, which will increase crop yields [45,46]. This study showed that the use of modified urea as urea + potassium humate and urea cocrystal, especially with high rates of 200 kg N ha$^{-1}$, had highly significant effects on all growth characteristics and yield attributes. These results are in accord with a previous study in 2018 [47]. It suggests that the mechanochemical synthesis of urea cocrystal (4urea.CaSO4) provides it with unique reactive properties towards the water as relative humidity and different reactive behavior from urea in pure water. Therefore, the release of nitrogen is balanced for the plant's requirements [47]. So, the use of modified urea fertilizers will be more effective in reducing nitrogen loss, increasing N availability to the plants, and enhancing it at the beginning of the experiment in soil [48]. Additionally, the humic compounds have a positive effect on plant growth that is commonly associated not only with the direct interaction of these molecules with the plant root and the activation of physiological processes in the plant but also through exhibiting multiple indirect effects [39,49]. In addition, humic compounds buffer pH, increase water retention, and mobilize nutrient availability. On the other hand, humic compounds may be promising remediation agents for degraded lands due to their ability to improve the soil's physical, chemical, and biological properties [49].

In this study, the use of modified urea fertilizers such as urea cocrystal and urea + potassium humate significantly affected silage yields compared to urea alone [17,50,51]. The urea cocrystal and urea + potassium humate treatments resulted in higher silage yields than urea treatments, especially with high N rates of 200 kg ha $^1$. The fresh and dry silage yields increased with the modified urea treatments due to the higher N uptake [52] (Table 8). Additionally, in response to using urea + potassium humate and urea cocrystal, there were increases in the corn grain yields by 16.88, 34.75, 24.69, and 40.28%, respectively, compared to the control. Urea treatments increased grain yields by 4.67% and 16.89%, respectively, compared to the control (Table 3). The N release rate can explain these findings from urea granules, which were higher than the urea cocrystal and urea + potassium humate in the beginning (until 3 weeks) of the corn growth [47], and that enhanced the nitrogen loss by leaching, indicated by the high concentration of mineral Nitrogen in the soil subsurface (30–60 cm) layer (Figure 5). While the release of nitrogen from urea cocrystal and urea + potassium humate increased after three weeks in most production systems, the grain development stage (starting at pollination) began about 75 to 95 days after planting [24].

In addition, the use of urea cocrystal had highly significant effects on the grain characteristics, including total N content, protein, and carbohydrates compared to the control, due to maintaining the balance of nitrogen release according to the plant requirements. Nitrogen levels also affect plant growth, leaf area production strength, and plant photosynthetic capacity. For example, the rate of photosynthesis in corn leaves decreases by reducing nitrogen levels [53]. Thus, grain yields, grain weight, and other components are significantly affected by nitrogen treatments [53]. A similar finding reported that the mixing of urea with gypsum increases its efficiency and is helpful for high-yielding aromatic rice varieties [26].

### 4.2. Mineral Nitrogen Transformations, Uptake, and Use Efficiency

The mineral nitrogen followed the same trend as the pre-studies without plants. Therefore, the release process was divided into three stages, starting with low concentration,

then an increase in concentration, then decreasing, and ending with concentration stability (Figure 4). The results showed that the concentration of ammonium, nitrate, and mineral nitrogen increased with time due to the decomposition of urea compound granules, which was enhanced by soil microbial activity [15]. The mineral nitrogen formed nitrate $NO_3^-$ and ammonium $NH_4^+$ from urea compound granules in the soil surface 0–30 cm and subsurface (30–60 cm) layers during the corn growth. The mineral nitrogen was specifically characterized as the nitrate form more often than ammonium, leading to increased nitrate leaching [54]. The pathways of N transformation are altered by the concentrations of N microbial species and the microbial population size [55]. The lower ammonium concentration was due to the low microbial biomass [55]. It also indicates a slight rate of nitrification where there is no population of nitrifying organisms [55]. Therefore, the ammonium released in the soil might be either captured by the microbial biomass, oxidized to nitrate, or volatilized. The mineral fertilizer concentration in the soil surface layer (0–30 cm) was significantly correlated with time for all the treatments except U200 and UPH100 because they did not change at a similar rate. These treatments (U200, and UPH100) approximately released an equivalent amount of nitrogen and had similar N uptake and grain yields during the experiment (Table 8). The mineral nitrogen concentration in the soil subsurface (30–60 cm) represented approximately 38–70% of the mineral nitrogen concentration in the surface soil (0–30 cm) layer during the corn growth period. This indicated a high level of nitrogen leaching from the surface layers, especially for the urea treatments with rapid nitrogen release at the beginning of the corn growth. The use of modified urea as urea + potassium humate and urea cocrystal with high rates of 200 kg N ha−1 showed highly significant effects on the N uptake in grain and stems and total nitrogen uptake by the corn crop compared to the control and urea alone (Figure 8). In addition, the treatments significantly affected the agronomic nitrogen use efficiency (ANUE) (Table 8). Urea + potassium humate and urea cocrystal improved the nitrogen use efficiency, especially at lower N rates (UC100 > UPH100 > UC200 > UHP200), while urea alone was not efficient, mainly when used in large quantities. Therefore, the optimal fertilizer application rate reduced the N rate, significantly increased N uptake in grains and dry matter, and improved ANUE as shown in Figure 8.

## 5. Conclusions

In this study, we show the unique effect of modified urea coated with potassium humate and synthesized with calcium sulfate on the soil mineral nitrogen release to increase its uptake and improve the use efficiency. The rapid release rates of conventional urea treatments at the beginning of the fertilization process compared to modified urea encourage increases in the chances of nitrogen loss through leaching to the subsurface layers, which is evident in the high content of the subsurface layer of mineral nitrogen. Therefore, the release rate of nitrogen is one of the most influential factors to reduce nitrogen loss and improve the nitrogen use efficiency of fertilizer. The release of nitrogen from any fertilizer depends on the reaction with water as relative humidity. Therefore, the coating of urea + potassium humate and the mechanochemical synthesis of urea cocrystal ($4urea.CaSO_4$) provides them with unique reactive properties towards the water as relative humidity and different reactive behavior from urea in pure water. This maintains the balance between the N soil content and the plant's basic needs. This study showed that the findings for fresh and dry matter yields and grain yields, grain characteristics and N uptake in grain and stems, and total nitrogen uptake and nitrogen use efficiency indicated that modified urea fertilizers such as urea + potassium humate and urea cocrystal were better than the conventional urea to improve corn yield productivity and N use efficiency.

**Author Contributions:** Conceptualization: S.S. and R.M.; Data curation, S.S. and D.A.; Formal analysis, R.M.; Investigation, D.A. and Z.B.; Methodology, D.A. and Z.B.; Supervision, R.M.; Validation, R.M.; Visualization, S.S.; Writing—original draft, S.S. All the authors commented on previous versions of the manuscript. All authors have read and agreed to the published version of the manuscript.

**Funding:** This research received no external funding.

**Institutional Review Board Statement:** Not applicable.

**Informed Consent Statement:** Not applicable.

**Acknowledgments:** The authors would like to thank everyone from the Rumokai experimental station of the Lithuanian Research Centre for Agriculture and Forestry for providing help during the trial, including field preparation, irrigation, and sampling. We also express our gratitude to AB Achema scientific experimental laboratory and Tadas Dambrauskas, Department of Silicate Technology, Faculty of Chemical Technology, Kaunas Technology University (KTU), Lithuania, for providing the fertilizers materials.

**Conflicts of Interest:** The authors declare no conflict of interest.

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
