# Peer review of "Influence of Modified Urea Compounds to Improve Nitrogen Use Efficiency under Corn Growth System"

_sustainability, doi:10.3390/su142114166_

Round 1

Reviewer 1 Report

1.The abstract was written incompletely,there are no some  key study contents such asmineral nitrogen transformation,nitrogen loss through leaching to the subsurface layers, nutrient utilization rate,

2.Some terms in the manuscript are confused and inconsistent,such as silage yields? Silage (leaves + straw) in Fig.2  and silage (leaves + stem) in 3.1 paragraph both are the same meaning,but better to use one.  Silage Fresh yield,Silage dry matter in Table 2。

The authors denote the treatments as T1, T2, T3, and control, U100, UPH200, UC200, etc. as well in the manuscript. So it is better to use one term between the two,

3. What is the mineral N unit in Table 5 and 6? what  do  four figures of between A and B,or C and D stand for in Fig. 5  and Fig 6 ?what does the word ‘straw’ in Table 8 and Fig.8  mean?

Reviewer 2 Report

Dear authors,

The work is very interesting for the entire scientific community. I ask you to technically process the paper based on the instructions for authors requested by the journal.

Best regards

Author Response

Thanks for your comment it was supportive and helpful.

About processing the article according to the author's instructions definitely, I will make sure that!

Reviewer 3 Report

Comments to authors:

In this study, the main goal was to study the effect of different modified urea compounds on corn yield and nitrogen utilization efficiency. However, the current manuscript is poorly written, and I have a hard time understanding the author's intentions in this manuscript. For example, the description of the experimental design in the Abstract section is unclear. The abstract does not report the important findings of the study in a clear manner and need to re-write. And there are many grammatical errors in this section as well as the full text. In the Introduction section, I did not find the novel description and research hypothesis of the study. The introduction does not clarify the importance of conducting this study and needs to be supplemented with the scientific issues and related advances in this study. In the Material and methods part, the description of the experimental design is not clear. Speak frankly, there are a lot of inappropriate sentences and illogical language in this manuscript, and unstandardized Figures and Tables in the Results section. Whats more, the reliable conclusion should be drawn under the field experiment for at least two years or more. Therefore, I do not think this manuscript can be published in this journal.

Round 2

Reviewer 3 Report

The authors have revised as requested and I have no further suggestions for the manuscript to be published.